# The Impact of Indoles Activating the Aryl Hydrocarbon Receptor on Androgen Receptor Activity in the 22Rv1 Prostate Cancer Cell Line

**DOI:** 10.3390/ijms24010502

**Published:** 2022-12-28

**Authors:** Eliška Zgarbová, Radim Vrzal

**Affiliations:** Department of Cell Biology and Genetics, Faculty of Science, Palacky University Olomouc, Slechtitelu 27, 783 71 Olomouc, Czech Republic

**Keywords:** prostate cancer, indoles, skatole, AR, AhR

## Abstract

The activation of the aryl hydrocarbon receptor (AhR) by xenobiotic compounds was demonstrated to result in the degradation of the androgen receptor (AR). Since prostate cancer is often dependent on AR, it has become a significant therapeutic target. As a result of the emerging concept of bacterial mimicry, we tested whether compounds with indole scaffolds capable of AhR activation have the potential to restrict AR activity in prostate cancer cells. Altogether, 22 indolic compounds were tested, and all of them activated AhR. However, only eight decreased DHT-induced AR luciferase activity. All indoles, which met the AhR-activating and AR-suppressing criteria, decreased the expression of DHT-inducible AR target genes, specifically *KLK3* and *FKBP5* mRNAs. The reduced AR binding to the *KLK3* promoter was confirmed by a chromatin immunoprecipitation (ChIP) assay. In addition, some indoles significantly decreased AR protein and mRNA level. By using CRISPR/Cas9 AhR knockout technology, no relationship between AhR and AR, measured as target gene expression, was observed. In conclusion, some indoles that activate AhR possess AR-inhibiting activity, which seems to be related to the downregulation of AR expression rather than to AR degradation alone. Moreover, there does not seem to be a clear relationship that would connect AhR activation with AR activity suppression in 22Rv1 cells.

## 1. Introduction

Prostate cancer is one of the most common types of cancer in men, and the risk of developing it increases with advanced age. More than 75% of patients with prostate cancer are men over the age of 65 [1]. According to Siegel’s statistic (2022), it is estimated that about 268,490 new cases of prostate cancer will be reported in the United States in 2022 [2]. In most deaths associated with prostate cancer, there is a predominant type known as castration-resistant prostate cancer (CRPC) [3]. Typical CRPC cellular mechanisms include androgen receptor (AR) overexpression, intratumoral synthesis of androgens, and the expression of shortened AR variants. [4]. These AR variants arise from alternative splicing of cryptic exons, and their activity is mostly ligand-independent [5]. The most abundant variant is AR-v7, which was indicated as a possible biological marker of prostate cancer development [6] since its levels have been reported to be significantly lower in normal prostate tissue than in prostate cancer tissue, or CRPC [5]. Prostate cancer with this splicing variant shows only a minor therapeutic response to commonly used anti-androgenic drugs (e.g., enzalutamide; ENZ) [7].

AR belongs to the nuclear receptor superfamily. Upon androgen (ligand) binding, AR translocates to the nucleus and binds to specific responsive elements in DNA. The most vital androgens are testosterone (T), dihydrotestosterone (DHT), and dehydroepiandrosterone (DHEA) [8]. Testosterone is the main male sex hormone and is converted to DHT by 5α-reductase enzyme. This process occurs in several target tissues, e.g., in the prostate [8,9]. Because both hormones bind to the AR, they are the main AR ligands, with DHT being more potent to AR than T [9]. AR activation through DHT is also essential for normal prostate development and function [10]. Imbalanced androgen secretion is linked with the occurrence of several associated diseases or syndromes, e.g., congenital lipoid adrenal hyperplasia, pseudohermaphroditism, or even affecting the development of prostate cancer [8,10,11]. Androgens modulate a wide range of biological responses in the human body through AR and AR became a significant therapeutic target in prostate cancer treatment [8]. Nowadays, the most common group of therapeutic drugs (i.e., direct approach) used in prostate cancer treatment are AR antagonists (e.g., bicalutamide, nilutamide, or ENZ) [12].

More than a decade ago, a different prostate cancer therapeutic strategy was suggested, assuming suppression of AR function through activation of the aryl hydrocarbon receptor (AhR), i.e., an indirect approach. Similarl to AR, upon ligand binding, AhR translocates into the nucleus, where it binds to a specific xenobiotic-responsive element in DNA, with a consequent increase in the expression of the target genes [13,14]. AhR has been primarily associated with biotransformation, but its ability to affect the function of other receptor pathways has also been described [15,16]. This receptor binds a wide number of endo- or exogenous molecules, with polyaromatic hydrocarbons (PAHs) or dioxins, such as, 2,3,7,8-tetrachlondibenzo-p-dioxin (TCDD), being the most potent exogenous ligands. Some PAHs have also been studied for anti-estrogenic and anti-androgenic effects [17]. 

An in vivo connection between AhR and AR in relation to prostate cancer was suggested in the study by Fritz et al. 2007, who used three different genotypes of mice, namely *AhR^+/+^*, *AhR^+/−^*, and *AhR^−/−^*, and studied the impact on prostate cancer development [18]. The obtained results suggested that the presence of AhR inhibits prostate carcinogenesis, and the model of prostate cancer development based on the genotype of the mice was as follows: *AhR^+/+^ < AhR^+/−^* < *AhR^−/−^*, where *AhR^+/+^* mice had the lowest incidence of prostate cancer development compared to *AhR^−/−^* mice (16% vs. 60%) [18]. 

Interestingly, only a limited number of studies have dealt with possible crosstalk between AhR and AR in human prostate cells. One such cross-talk can represent AhR-mediated AR degradation, since AhR has been demonstrated as a ligand-dependent E3 ubiquitin ligase that induces proteasomal degradation of AR in the androgen-sensitive prostate cancer cell line LNCaP [19]. However, it appears that the effect differs between cell lines since the castration-resistant type (C4-2 cell line), TCDD did not induce AR degradation through AhR activation [20]. The AhR and AR protein levels were significantly decreased after the treatment of LNCaP cells with another potent AhR agonist from the PAH group, 3-methylcholanthrene (3MC) [21]. The anti-androgenic effects of AhR ligands were also described for various AhR agonists, such as chrysene, benzo[k]fluoranthene, and benzoapyrene (B[a]P) but not for anthracene or pyrene. Moreover, listed AhR agonists with anti-androgenic effects also increased c-fos and c-jun mRNA levels [17].

The study of Arabnezhad et al. 2020 considered the effects of endogenously activated AhR on AR [22]. The prostate cell line LNCaP was treated with 6-formylindolo [3,2-b]carbazole (FICZ) in the presence or absence of the AR ligand testosterone. After treatment, mRNA (*AR*, *KLK2*, *TMPRSS2,* and *PSA*), PSA protein levels, and DHT levels were evaluated as the end points. The study confirmed that FICZ induced *CYP1A1* activity, which is a marker of AhR activation. In addition, a significant decrease in AR-target gene mRNA expression was observed with the combination of FICZ (50 nM) and testosterone (100 nM). PSA protein and DHT levels were reduced after treatment with FICZ + T. However, no decrease was observed in the absence of testosterone. The obtained results indicated that AhR plays an important role in AR signaling and that FICZ has anti-androgenic effects through the AhR/AR pathway [22]. Moreover, Morrow et al. 2004 study suggested that the stability of AR protein is strictly dependent on AhR ligand and that AhR/AR inhibitory crosstalk is more likely promoter specific [23]. Using a co-immunoprecipitation assay, it was demonstrated that AhR forms a complex with AR and that this process is fundamentally facilitated by DHT. Furthermore, the presence of DHT decreased 3MC-induced transcription of *CYP1A1* [21].

Interestingly, the concept of AhR-mediated degradation of AR was described for other compounds, namely Icaritin [24] and Carbidopa [25]. Icaritin is one of the major metabolites of the compound Icariin [26,27]. Both of these substances are naturally occurring polyphenols, which can be found in plants of the genus *Epimedium* [27]. Carbidopa is a decarboxylase inhibitor used to treat Parkinson’s disease [28]. Both of these compounds have been shown to induce AhR activation with consequent degradation of AR [24,25]. However, while Icaritin induced this effect in LNCaP, C4-2 and 22Rv1 cells, Carbidopa was demonstrated to have such an effect in LNCaP cells only. Moreover, these compounds suppressed the tumor proliferation of LNCaP implanted cells in nude mice [24,25].

Therefore, available studies strongly indicate the presence of AhR/AR crosstalk in prostate cancer cells, as various AhR ligands were reported to inhibit prostate cancer development, probably through several different mechanisms [22]. Published data suggest that only strong and potent AhR ligands are capable of inducing AR degradation through AhR activation [19].

Recently, our research team observed that the group of mostly synthetic compounds with an indole scaffold has the ability to activate AhR in hepatocarcinoma cells (AZ-AhR, derived from the parental HepG2 cell line) [29] with efficacy comparable to TCDD (5 nM).

Therefore, the objective of this study was to determine whether these indoles have the ability to suppress AR function through AhR activation and thus slow the proliferation of cancer cells.

## 2. Results

### 2.1. The Effects of Indoles on AhR and AR Transcription Activity

Firstly, the new reporter cell line 22AhRv1 was constructed, characterized (Appendix A), and subsequently used to monitor AhR transcription activity in a prostate-specific environment. The AIZ-AR reporter cell line [30] was used to monitor AR transcription activity in the absence (agonist mode) or presence (antagonist mode) of DHT. 

In the 22AhRv1 reporter cell line, the ability of tested indoles to activate AhR was measured after 4 h and 24 h. Significant AhR activation was observed for 1MI, 2MI, 4MI, 5MI, 6MI, 7MI, 2,5DMI, 6MeO, 4MeO1MI, and 7MeO4MI after 4 h (Figure 1A). Interestingly, a positive control, TCDD, did not significantly activate AhR above the level of UT after 4 h, despite significant activation after 24 h (Appendix A and Figure 1B). After 24 h, all indoles significantly and dose-dependently activated AhR with efficacy comparable to TCDD (Figure 1B). An exception to this rule was 3MI (skatole), which had no effect at the highest concentration.

In the AIZ-AR reporter cell line, the ability of the tested indoles to affect AR was measured after 24 h. DHT-inducible AR-dependent luciferase activity reached approximately 7-fold induction next to UT (Figure 2A). True AhR ligands, TCDD and FICZ, were tested as well. While TCDD had no effect, FICZ significantly induced AR-mediated luciferase activity. Furthermore, the anti-androgen ENZ significantly decreased this activity. This was probably due to the use of regular serum in the medium, which contains certain levels of androgens. Of all the compounds tested, only 7MeO markedly and significantly increased AR activity (approximately 2.5 times). Interestingly, a concentration-dependent decrease in luciferase activity was observed for 3MI (with an IC_50_ approx. 10 µM).

In antagonist mode, a positive control ENZ significantly inhibited DHT-inducible AR-dependent luciferase activity to the level of UT. Interestingly, TCDD decreased luciferase activity by approximately 30%, while FICZ increased it above DHT by approximately 20%. This finding suggests that the impact of AhR activation on AR activity may be strongly dependent on the nature of the AhR ligand. A significant concentration-dependent decrease in AR activation was observed for 3MI, 4MI, 1,3DMI, 2,3DMI, 2,3,7TMI, 4,6DMI, 5,6DMI, and 7MeO4MI (Figure 2B). A decrease in AR activity was most remarkable for 3MI (with IC_50_~5 µM) and 4MI (IC_50_~50 µM). Some compounds, such as 2,5DMI, 2,3,3TMI, 7MeO, or 5MeO2MI, were able to coactivate AR in combination with DHT (10 nM), but this was consistent with their activity observed in the agonist mode (Figure 2A). Compounds that were able to significantly activate AhR and simultaneously inhibit DHT-activated AR were used for further experiments (Table 1).

Due to the decline in luciferase activity, we performed MTT and Crystal violet assays. After 24 h of incubation, most of the substances (concentration range 1–100 µM) have been described as nontoxic by both assays, since the decrease in cell viability was not less than 20% (Appendix A). However, an enormous decline was observed for 3MI and partially for 4MI by means of Crystal violet assay only (Appendix A). We measured a decrease of 39% and 60% of untreated cells for 3MI and 4MI, respectively. A calculated inhibitory concentration for 3MI was estimated as IC_50_~60 µM.

### 2.2. Effects of Selected Indoles on the Target Gene Expression

All tested indoles increased concentration-dependent *CYP1A1* mRNA expression, which is a marker of AhR activation [14]. The positive control, TCDD (10 nM), increased *CYP1A1* expression by 106-fold (Figure 3A). The weakest inducers of *CYP1A1* were 4,6DMI and 5,6DMI, while 4MI was the strongest. Another AhR target gene, *AhRR*, was induced by TCDD, and only 4MI, 1,3DMI, and 7MeO4MI showed weak induction (Figure 3B). Interestingly, 3 MI decreased *AhRR* mRNA levels.

All tested indoles significantly reduced the DHT-inducible expression of AR target genes, namely *KLK3* (Figure 3C) and *FKBP5* (Figure 3D). Induction by DHT was approx. 1.5-fold and 4-fold for *KLK3* and *FKBP5*, respectively. The expression of the *UBE2C* gene, which is regulated by AR-v7 only [31], was mildly decreased by all indoles (Figure 3E). The strongest decrease (up to 60% of control cells) was observed for 3MI.

### 2.3. Binding of AR to the KLK3 Promoter after Treatment with Selected Indoles

In order to demonstrate whether the impact of the tested indoles started at the transcription level, we performed the ChIP assay. 22Rv1 cells were incubated with tested indoles (10 µM) and controls in antagonist mode (with DHT, 10 nM) for 90 min. The positive control DHT (10 nM) enriched the *KLK3* promoter with AR by 2.8–4.9 fold (Figure 4A). The presence of anti-androgen ENZ resulted in a significant reduction compared to DHT. Well known AhR ligands, TCDD and FICZ, were able to decrease a DHT-stimulated AR enrichment. A mild decrease has been observed for all tested indoles. The obtained data suggest that the observed decrease in *KLK3* and *FKBP5* mRNAs started already at the level of binding of AR to DNA, thus affecting AR functionality.

### 2.4. Effects of Selected Indoles on AR-fl and AR-v7 Protein Levels

In the context of the proposed hypothesis, we evaluated the AR protein level. A significant dose-dependent decrease was observed for both variants, AR-fl (110 kDa, Figure 5A,C) and AR-v7 (89 kDa, Figure 5B,C), exposed to 3MI, 4MI, 2,3,7TMI, and 7MeO4MI. Interestingly, 4MI in 1 µM increased the level of DHT-inducible AR-v7 protein by up to 200%. In order to verify the change in either degradation or transcription, we measured mRNA levels. And surprisingly, the mRNA levels of both AR variants decreased in a similar pattern, with the strongest effect shown by 3MI (IC_50_ approx. 10 µM), followed by 4MI, 2,3DMI, 2,3,7TMI, and 7MeO4MI (Figure 5D,E). Thus, the tested indoles probably induce downregulation in AR mRNA, which is further reflected in a decrease of AR proteins.

### 2.5. CRISPR/Cas9 AhR Knockout

Finally, we tried to evaluate the connection between AhR, AR, and the strongest AhR activator in 22Rv1 cells, 4 MI, by genetic knockout. For this purpose, 22Rv1 cells were transiently transfected with the CRISPR/Cas9 plasmid for AhR-KO. In all experiments, a decrease in *AhR* mRNA was successfully observed between WT and KO samples (Figure 6A). *CYP1A1* mRNA expression in AhR KO cells decreased slightly after treatment with TCDD or 4MI in combination with or without DHT (Figure 6B). However, the combined treatment of 4MI and DHT did not show any recurrent trend in *KLK3* mRNA level (Figure 6C) and had no impact on *FKBP5* mRNA level (Figure 6D) when comparing WT and AhR KO cells. The effect on *AR* mRNA levels is given in Appendix A.

## 3. Discussion

The main goal of this study was to demonstrate if AhR activation by certain indoles can result in the degradation or at least suppression of AR activity in prostate cancer cells. This assumption was based on the observations that strong AhR ligands degraded AR in the LNCaP prostate cell line [15,19] and that two other compounds, namely Icaritin [24] and Carbidopa [25], which could activate AhR, suppressed prostate cancer via AhR-mediated degradation of AR. Effects of these compounds and indoles from our study on AhR and AR pathways are summarized in Table 2. The tested indoles were selected on the basis of a recent report by our research group, in which methyl- and methoxyindoles were identified as AhR agonists [29]. In the human hepatocarcinoma AZ-AhR cell line, all indoles activated AhR to some extent, and the strongest ones were reported to be 4 MI, 5 MI, 6 MI, 2,5 DMI, and 7MeO [29]. Similarly, all tested indoles activated AhR in the 22Rv1 prostate cell line after 24 h with efficacy comparable to TCDD, which drastically less activated AhR (5-fold, 10 nM) than AZ-AhR (1000-fold, 5 nM). A partial explanation of such a huge difference lies in the significantly lower amount of AhR protein in the prostate (22Rv1) than in AZ-AhR cells (personal observation).

Despite the fact that all indoles activated AhR comparably to TCDD (Figure 1), only some of them were able to suppress AR activity (Figure 2) and consequently AR-target gene expression (Figure 3). Interestingly, typical high affinity AhR ligands, TCDD and FICZ, had a opposite impact on the DHT-inducible AR-mediated luciferase activity (Figure 2), despite concentration-dependent induction of AhR-mediated luciferase activity (Appendix A). Furthermore, selected indoles that suppressed AR activity inhibited DHT-inducible AR binding to the *KLK3* promoter to some extent, suggesting an impact on the initial transcription event. Additionally, these indoles suppressed AR mRNA levels, which were consequently reflected in AR protein levels. This suggests a different type of mechanism from that initially expected. Additionally, despite very strong AhR activation next to TCDD, most of the indoles have not been demonstrated to be true AhR ligands so far, and thus, the lack of AR degradation effect may stand beyond this fact. This is consistent with the original observation by Ohtake et al., 2007 [19], who found that only full agonists such as TCDD, BaP, and 3MC could degrade AR but weaker ligands (e.g., indirubin–indole scaffold, β-naphthoflavone) could not. 

Further, next to the nature of a ligand/agonist of AhR, probably a specific environment plays a role since the study by Ohtake et al., 2007 [19] used LNCaP cells treated for 3 h in medium with 0.2% charcoal-stripped serum, while we used 22Rv1 cells in medium with 10% regular serum and added 10 nM DHT to mimic basal androgen secretion in the human body. Another difference between these two cell lines lies in the presence of AR variants, where LNCaP has one (AR-fl), while 22Rv1 has two (AR-fl, AR-v7) [5]. Furthermore, AR-v7 displays nuclear localization even in the absence of androgen and acts as a transcription factor [5]. 

Additionally, it seems that different types of prostate cancer cells have shown different sensitivity to affecting the AR pathways through AhR activation. The AR degradation process through AhR activation was not identified in all types of prostate cancer cells. In the androgen-sensitive LNCaP cell line, which was isolated from the left supraclavicular lymph [32], the mechanism of AR degradation through AhR activation by a strong ligand has been observed multiple times [20,22]. LNCaP cells are classified as AR and AhR-positive [21]. However, the same AhR ligand (TCDD) did not induce AR degradation in the castration-resistant C4-2 cell line [20], which was derived from the parental LNCaP [33]. This may be due to the fact that different types of prostate cancer are diverse in AhR and AR receptor content, and subcellular localization, and that AhR presence may affect AR phosphorylation status [34]. This is probably also reflected in our data, since ChIP results demonstrated a decrease in AR binding to the *KLK3* promoter after 90 min (Figure 4), while AR protein levels (Figure 5) were almost unaffected by most of the tested indoles after 24 h with the 10 µM concentrations used in ChIP. Thus, our results suggest a mix of potential post-translational modification of AR or affected interaction with other transcription partners. This consequently leads to a decrease in AR binding to the *KLK3* promoter and a decrease in the mRNA levels of AR-target genes.

Probably the most interesting finding is the impact of 3MI (skatole) on AR activity and cell viability. Skatole is naturally a product of the intestinal microbiota and can be found in wide range of concentrations in the serum at least of patients with hepatic encephalopathy [35]. Unfortunately, it is not known what the average concentration of skatole in the serum of healthy individuals is. Therefore, our results may suggest a hypothesis that men with detectable skatole concentration in the serum are less likely to develop/have AR-dependent prostate cancer. The proposition of this hypothesis is based on the observation of a decline in all AR-mediated parameters we studied, as well as on the strongest decline in the Crystal violet assay, which is a reflection of the proliferative capacity of the cells. 

Furthermore, we observed a visible morphological change in 22Rv1 cells at the end of skatole treatment (personal observation). Thus, it is likely that skatole triggers some sort of apoptotic event and apparently has more molecular targets inside the cells. Our assumption is based on the observation of different skatole action in relation to other indoles as it reduced rather than induced *AhRR* mRNA (Figure 3B) and consistently induced concentration-dependently *AhR* mRNA (personal observation). The expression of *AhRR* and *AhR* is regulated by NF-kB [36], and the combination of genotoxic action, demonstrated for skatole in bronchial cells [37] (that is, activation of p53), and activation of NF-kB could explain these differences. However, this must be revealed in future research as a possible relationship (if any) between skatole blood level and prostate cancer incidence in men.

We believe that skatole deserves further investigation in the future in relation to prostate cancer.

The hypothesis of AR degradation through AhR activation by selected indolic compounds was tested. All 22 tested indoles displayed promising AhR-activating potential in the newly developed 22AhRv1 reporter cell line. However, only 8 of them were able to supress DHT-induced AR activation, namely 3MI, 4MI, 1,3DMI, 2,3DMI, 2,3,7TMI, 4,6DMI, 5,6DMI, and 7MeO4MI. At the mRNA level, 8 selected indoles increased *CYP1A1* expression and significantly reduced DHT-inducible *KLK3* and *FKBP5* expression. Moreover, following ChIP analysis revealed reduced DHT-inducible binding of AR to the *KLK3* promoter by selected indoles. The effect of the strongest AhR activator in this study, 4MI, was further monitored using transient transfection with the CRISPR/Cas9 AhR KO plasmid. However, after AhR KO, no change in AR-targets was observed. To conclude, some tested indoles showed the ability to suppress DHT-induced AR activation; however, this phenomenon does not seem to be associated with AhR activation. Therefore, the hypothesis of AR degradation via AhR activation for selected indoles has not been confirmed.

## 4. Materials and Methods

### 4.1. Cell Lines

Human prostate carcinoma epithelial cell lines 22Rv1 (ECACC No. 105092802) and AIZ-AR [30] were cultured in Rosewell Park Memorial Institute (RPMI) 1640 medium supplemented with 10% fetal bovine serum (FBS), 1% non-essential amino acids, and 2 mM L-glutamine (Sigma-Aldrich, Missouri, USA). At the beginning of the experiments, GENERI BIOTECH s.r.o. (Hradec Králové, Czech Republic) performed the authentication of the 22Rv1 cell line. Cells were maintained in a humidified incubator at 37 °C and 5% CO_2_. Cell lines were regularly tested for the presence of mycoplasma using the MycoAlert^TM^ Mycoplasma Detection Kit (Lonza, Basilej, Switzerland). 

### 4.2. Compounds and Reagents

The tested indoles were selected based on previous research [29] in our laboratory and are listed in Table 3. The table contains key structures of tested indoles is given in Appendix A. The chemicals used are listed in Appendix A. 

### 4.3. Reporter Gene Assay (RGA)

Initially, a cellular system was established to monitor AhR transcription activity in the 22Rv1 cell line (Appendix A). AIZ-AR reporter cell line [30] was used for evaluation of AR transcription activity. Cells were seeded in 96-well plates at a density of 2 × 10^4^ cells per well in a volume of 200 µL and stabilized overnight. 22AhRv1 cells were incubated with tested indoles for 4 h and 24 h. AIZ-AR cells were treated with indoles for 24 h in agonist and antagonist mode (in the presence of DHT 10 nM). DMSO (untreated; 0.1%; *v*/*v*) was used as the vehicle/negative control, TCDD (10 nM) was used as the positive control in 22AhRv1 and DHT (10 nM) was used as the AIZ-AR positive control. Furthermore, anti-androgen ENZ (10 µM) was used as AR suppression control in AIZ-AR cells, and the effects of TCDD (10 nM) and FICZ (10 µM) in the presence or absence of DHT (10 nM) were also evaluated. After treatments, cells were lysed with 1x Reporter Lysis Buffer (Promega, Wisconsin, USA) and luciferase activity was measured with the Infinite M200.

### 4.4. Quantitative Reverse Transcriptase PCR (RT-qPCR)

Only indoles capable of activating AhR and simultaneously suppressing AR activity were selected for PCR analysis. Cells were seeded in 6-well plates at a density of 1.5 × 10^6^ cells per well in a volume of 1.5 mL and stabilized overnight. Cells were treated with 8 selected indoles in 4 concentrations (1, 10, 50, and 100 µM) in antagonist conditions (with DHT at 10 nM). After 24 h of treatment, cells were washed with PBS, and total mRNA was isolated using TRI Reagent^®^ (Molecular Research Centre, Ohio, USA), according to protocol. Quality was monitored in the A260/A280 ratio. cDNA was synthesized from 1000 ng of total RNA using M-MuLV Reverse Transctiptase (New England BioLabs, Massachusetts, USA) at 42 °C for 60 min in the presence of Random Primer 6 (100 pmol/µL; New England BioLabs). Subsequently, enzyme activity was inactivated when incubated at 65 °C for 10 min in Dry Bath Incubator (Major Science, California, USA). The mRNA expression was evaluated with KiCqStart^®^ Probe Assays (Sigma-Aldrich) or with SYBR^®^ Green (Roche Diagnostics, Prague, Czech Republic) according to the manufacturer’s recommendations. The expression of genes related to the AhR pathway (*AhR, AhRR, CYP1A1*) and AR-target genes (*AR-fl, AR-v7, FKBP5, KLK3, UBE2C*) was monitored. The list of used probes and primers is given in Table 4. qRT-PCR was performed on the Light Cycler 480 II apparatus (Roche Diagnostic), and the reaction settings for KiCqStart^®^ Probe Assays analysis are shown in Table 5. The reaction by SYBR^®^ Green analysis was performed according to the manufacturer´s recommendations. Sample measurements, including the non-template control (NTC), were performed in triplicate, and gene expression was normalized to the housekeeping gene *GAPDH*. The data obtained were evaluated by the delta-delta method.

### 4.5. Western Blot

Cells were seeded in 6-well plates at a density of 1.5 × 10^6^ cells per well in a 1.5 mL volume and stabilized. The treatment was similar to that for RT-qPCR. Total protein extracts were isolated for each sample after 24 h of treatment. Cells were washed with 1× PBS, scraped into 1 mL of PBS (cold) and centrifuged (4000 rpm/1500 rcf, 3 min, 4 °C). Pellet was re-suspended in 80 µL protein lysis buffer (pH 7.5; 50 mM HEPES, 5 mM EDTA, 150 mM NaCl, 1% Triton X-100 with anti-protease and anti-phosphatase cocktail) and centrifuged at 13 000 rpm (14,500 rcf), 15 min, 4 °C. The supernatant was collected, and the protein concentration in each sample was determined by 1x Bradford reagent (Sigma Aldrich). Protein samples were separated by sodium dodecyl sulfate-polyacrylamide gel electrophoresis (SDS-PAGE; 10% running gel, 4% stacking gel) in the BioRad Miniprotean system (California, USA), according to the manufacturer’s recommendations. SDS-PAGE was followed by semi-dry protein transfer onto a polyvinylidene difluoride (PVDF) membrane. The membranes were stained with Ponceau S Rouge solution (0.1% *w/v* in 5% acetic acid), washed with 1xTBS/Tween, and incubated with 5% non-fat dried milk for 1 h at room temperature (RT). Membranes were incubated overnight with appropriate antibody diluted in 5% solution of bovine serum albumin (BSA) in 1xTBS/Tween at 4 °C. After incubation, the membranes were washed with 1xTBS/Tween and incubated with secondary horseradish peroxidase-conjugated antibody diluted in 5% non-fat dried milk for 1 h at RT. List of used antibodies and their concentrations is given in Table 6. Membranes were analyzed using WesternSure^®^ PREMIUM Chemiluminiscent Substrate (Li-Cor, Nebraska USA) and the C-DiGit Chemiluminiscence Western Blot Scanner (Li-Cor). Obtained results were processed by Image Studio 5.0 for C-DiGit Scanner software. Protein expression was normalized per β-actin (43 kDa).

### 4.6. Chromatin Immunoprecipitation (ChIP)

22Rv1 cells were seeded in a 60-mm plate at a density of 4 × 10^6^ cells per plate in a 4 mL volume and stabilized overnight. The following day, cells were incubated with selected indoles in antagonist mode (only 10 µM concentration with DHT 10 nM) and controls (DMSO in a ratio of 1:1000; DHT 10 nM). Furthermore, the effect of some other compounds (TCDD 10 nM; FICZ 10 µM; ENZ 10 µM) in combination with DHT 10 nM was evaluated. Cells were incubated with the tested compounds for 90 min at 37 °C. Subsequently, ChIP was performed according to the manufacturer’s recommendations in the manual for the Simple ChIP Plus Enzymatic Chromatin IP Kit (Cell Signaling Technology, Danvers, MA, USA) as previously described, with minor modifications [29]. Briefly, an aliquot of 15 µg of digested chromatin was resuspended in ChIP buffer in total volume of 500 µL for each sample and incubated overnight with ChIP validated antibody (4 µL of anti-AR antibody; Cell Signalling Technology). Also, a control sample with normal rabbit IgG was prepared (1 µL of IgG antibody). The following day, 25 µL of Protein G magnetic beads were added to each sample and incubated with rotation for 45 min at 4 °C. The beads were separated in the Magnetic Separation Rack, and subsequently several washing steps (using low- and high-salt buffer, according to the manufacturer) were performed. Then, 150 µL of ChIP elution buffer was added to each sample. The elution was achieved with incubation at 65 °C for 30 min with shaking (1200 rpm/100 rcf). Magnetic beads were separated, and eluted chromatin was transferred into a clear tube. After that, 6 µL of NaCl (5 M) and 2 µL of proteinase K were added to each sample and incubated at 65 °C for 15 min with shaking (1200 rpm/100 rcf). Thus, 750 µL of DNA binding buffer were added to each sample, which was briefly vortexed and centrifuged. Samples were transferred to spin columns and washed with 600 µL of DNA wash buffer. Chromatin was eluted from columns by adding 45 µL of DNA elution buffer and collected in new tubes. The result analysis was performed using RT-qPCR in Light Cycler 480. For each well, the reaction mix contained 2 µL of eluted ChIP sample, 5 µL SYBR Green (Roche), 2 µL of PCR grade H_2_O (Roche) and 1 µL of *KLK3* Promoter Primers (SimpleChIP, Cell Signaling). The setting of this reaction is given in Table 7. For the purpose of agarose gel analysis, the next PCR run was performed with elongation for 30 cycles only. The product of such a reaction was analyzed using agarose electrophoresis (4% gel, 65 V, 1.5 h). Samples were dyed with GelRed^®^ nucleic acid stain (Biotium, California, USA). The result was visualized with the Syngene G:BOX gel documentation system (Cambridge, UK).

### 4.7. CRISPR/Cas9 AhR Knock-Out

To verify the relationship between AhR and AR, 22Rv1 cells were transiently transfected with an AhR CRISPR/Cas9 KO plasmid according to the manufacturer’s protocol (Santa cruz Biotechnology, Texas, USA), with minor modifications. Briefly, 22Rv1 cells were seeded in a 6-well plate (2 × 10^5^ cells and 3 mL of medium per well) and stabilized for 24 h in a humidified incubator (80% confluence). Subsequently, cells were transfected with the CRISPR/Cas9 AhR KO plasmid or with the control CRISPR/Cas9 plasmid, as a negative control. Solution A, 1 µg of plasmid DNA in Plasmid Transfection Medium in a total volume of 150 µL, and solution B, 5 µL of UltraCruz^®^ Transfection Reagent in Plasmid Transfection Medium in a total volume of 150 µL, were prepared for each well and let stand for 5 min at RT. Subsequently, solution A was added to solution B, vortexed, and incubated for 20 min at RT. 

Plates with stabilized cells were transferred to the flow box, and the growth medium was replaced by a fresh one. To each well, 300 µL of transfection solution A + B was added dropwise. The plate was gently mixed by swirling, and cells were incubated with transfection reagents for 48 h. After incubation, the transfection process was visually confirmed by fluorescence microscopy by detection of green fluorescence protein (GFP). Subsequently, transiently transfected cells were exposed to the tested compounds (UT, DHT 10 nM, TCDD 10 nM with or without DHT 10 nM, 4 MI 100 µM with or without DHT 10 nM) for 24 h. Total mRNA was isolated, and the expression of genes (*AhR*, *CYP1A1*, *AR-fl*, *KLK3*, and *FKBP5*) was evaluated as described earlier (Section 4.4). For evaluating *AhR* mRNA levels, specific primers were designed for exon 5, which is the target DNA sequence for the CRISPR/Cas9 AhR KO plasmid.

### 4.8. Statistical Analysis

The obtained data were processed using GraphPad Prism Version 9.4.1 (GraphPad Software, San Diego, CA, USA). Significant values in all experiments were determined by two-way ANOVA (symbol * in the charts). Western blot results were analysed using Image Studio 5.0 for C-DiGit Scanner software (Li-cor, Nebraska, USA).

## Figures and Tables

**Figure 1 ijms-24-00502-f001:**
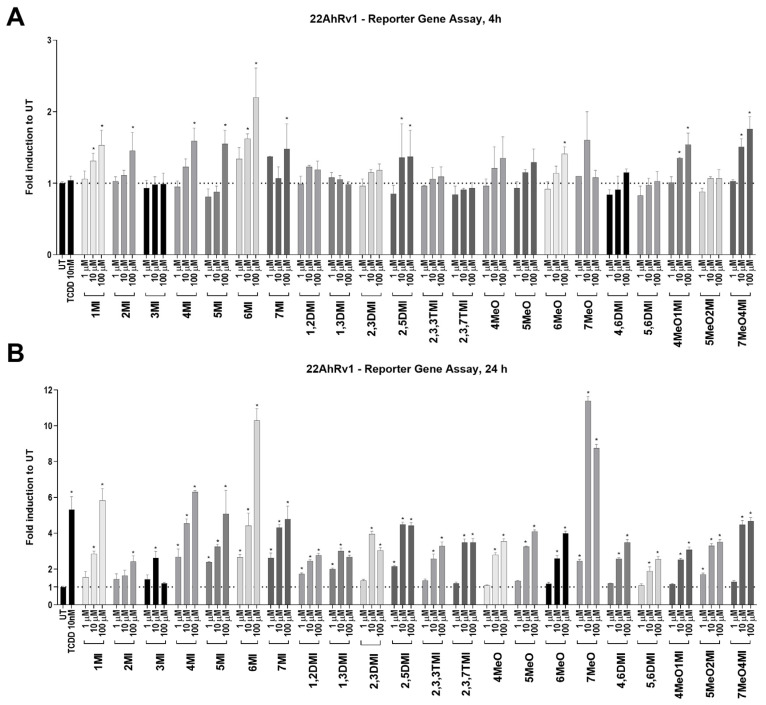
AhR transcription activity after indoles treatment. 22AhRv1 cells were incubated for 4 h (**A**) and 24 h (**B**) with indoles in the concentration range of 1 µM to 100 µM or controls (UT; TCDD 10 nM). The transcription activity of AhR was measured using RGA. The data obtained are averages of three to five experiments and are expressed as fold induction over untreated (UT; DMSO-treated) cells. * represents a significant difference (*p* < 0.05) between untreated (UT) and compound-treated cells.

**Figure 2 ijms-24-00502-f002:**
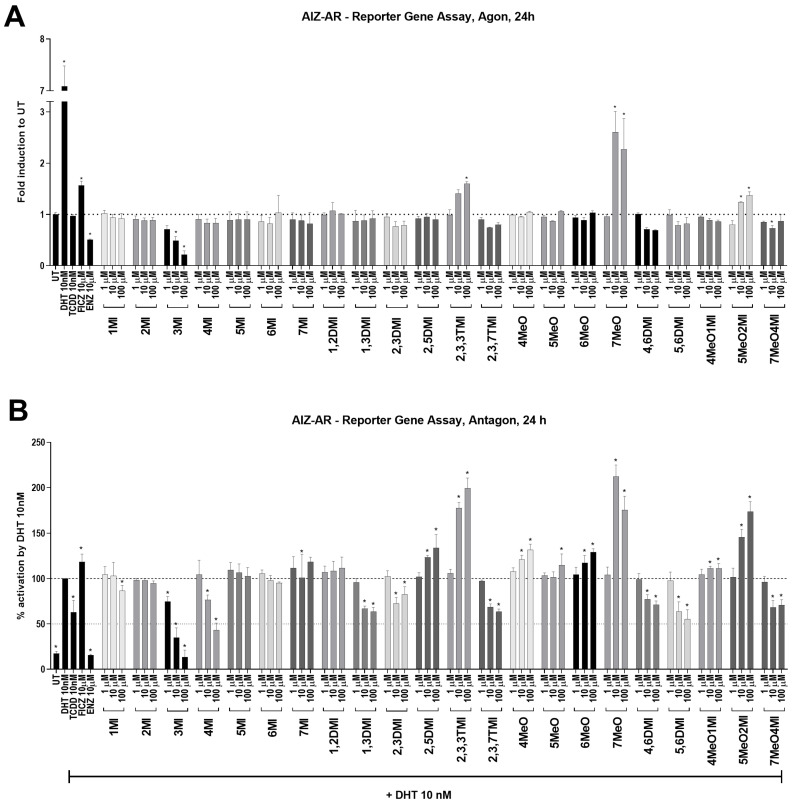
AR transcription activity after indoles treatment. AIZ-AR cells were incubated for 24 h with indoles and controls in the absence (**A**) or presence of 10 nM DHT (**B**). The transcription activity of AR was evaluated using RGA. In agonist mode (**A**), the data obtained are expressed as fold inductions to UT. In antagonist mode (**B**), the results are expressed as a percentage, when DHT 10 nM is set to 100%. * represents a significant difference (*p* < 0.05) between untreated (UT) and compound-treated cells.

**Figure 3 ijms-24-00502-f003:**
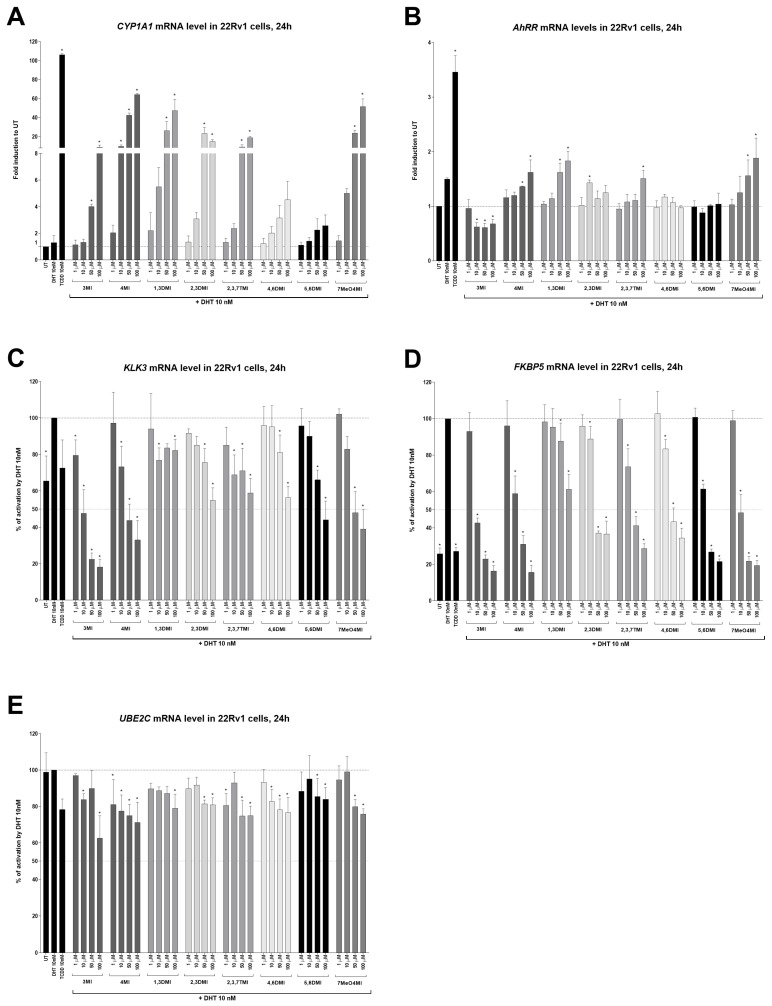
Effects of indoles on the expression of AhR- and AR-target genes. 22Rv1 cells were incubated with indoles (1 µM to 100 µM) in the presence of DHT 10 nM or controls (UT; DHT 10 nM; TCDD 10 nM) for 24 h. The induction of AhR-target genes *CYP1A1* (**A**), *AhRR* (**B**) and AR-target genes *KLK3* (**C**), *FKPB5* (**D**), and *UBE2C* (**E**) were determined by RT-qPCR. The data obtained were normalized per the housekeeping gene *GAPDH* levels. The results are expressed as fold induction in untreated (UT; DMSO-treated) cells (for AhR-target genes) or as a percentage when DHT 10 nM is set to 100% (for AR-target genes). * represents a significant difference (*p* < 0.05) between untreated (UT) and compound-treated cells (**A**,**B**) or DHT and compound + DHT-treated cells (**C**–**E**).

**Figure 4 ijms-24-00502-f004:**
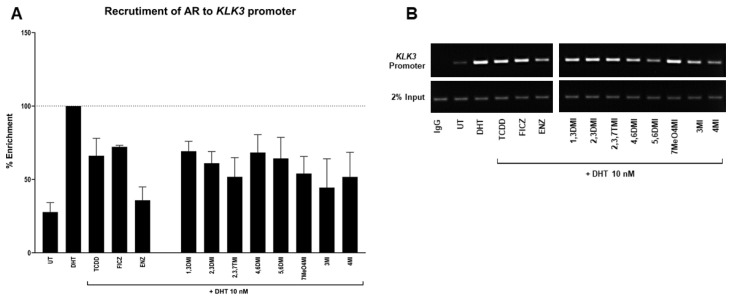
Chromatin immunoprecipitation. 22Rv1 cells were incubated with the tested compounds for 90 min (all indoles, FICZ, and ENZ at 10 µM; DHT and TCDD at 10 nM). Thereafter, the enrichment of the *KLK3* promoter with AR was evaluated. The results are expressed as a percentage of positive control. Enrichment of *KLK3* promoter by DHT was set to 100% (**A**). Representative DNA fragments were amplified using PCR and analyzed with agarose gel electrophoresis (**B**).

**Figure 5 ijms-24-00502-f005:**
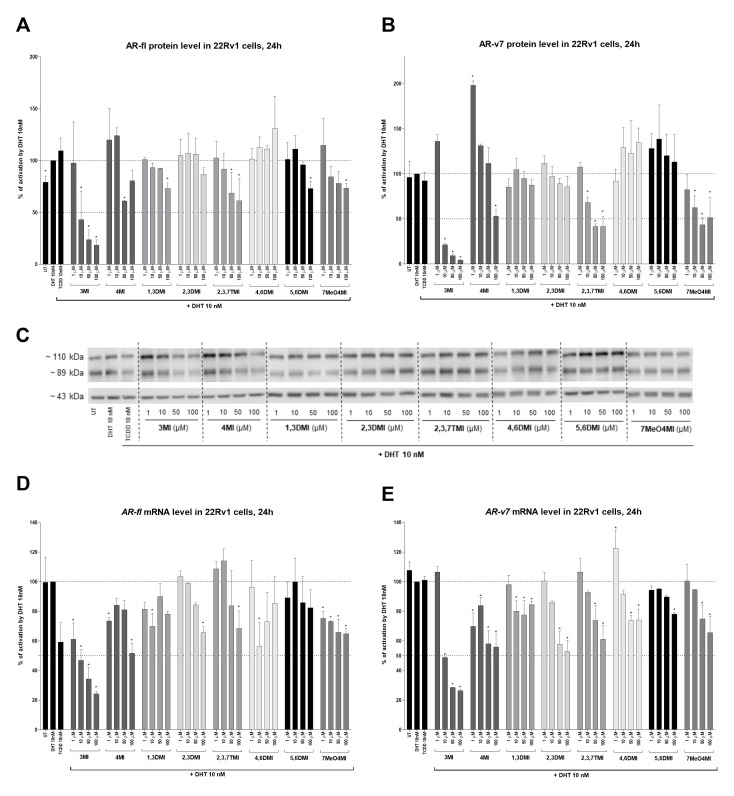
Effects of indoles on AR-fl and AR-v7 protein levels. 22Rv1 cells were incubated with indoles (1 µM to 100 µM) in the presence of 10 nM DHT or controls (UT; TCDD 10 nM; DHT 10 nM) for 24 h. The protein levels of AR-fl (110 kDa; **A**) and AR-v7 (89 kDa; **B**) were evaluated using semi-dry western blotting. The results obtained were normalized per β-actin (43 kDa). A representative blot is shown in (**C**). For comparison, *AR-fl* (**D**) and *AR-v7* (**E**) mRNA levels were also determined using RT-qPCR. The data obtained were normalized per the housekeeping gene *GAPDH*. The results are expressed in percentages, when DHT 10 nM is set to 100%. * represents a significant difference (*p* < 0.05) between DHT and compound + DHT-treated cells (**C**–**E**).

**Figure 6 ijms-24-00502-f006:**
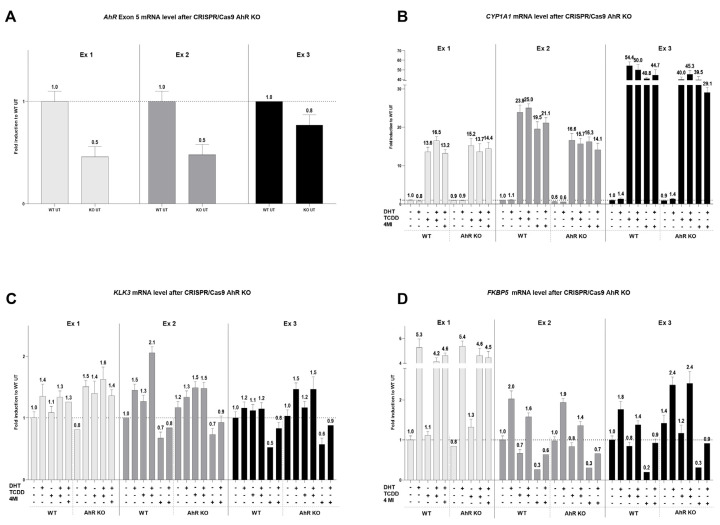
Expression of target genes after CRISPR/Cas9 AhR knockout. 22Rv1 cells were transiently transfected with the CRISPR/Cas9 AhR knockout plasmid or the Control CRISPR/Cas9 plasmid. After 48 h, transfected cells were treated with controls (UT; TCDD 10 nM with or without DHT 10 nM; DHT 10 nM) and 4MI 100 µM with or without DHT 10 nM. The induction of *AhR* (**A**), *CYP1A1* (**B**), *KLK3* (**C**), and *FKBP5* (**D**) was determined by RT-qPCR. The data obtained were normalized per the housekeeping gene *GAPDH*. The results are expressed as fold induction in wild-type UT. In total, three independent experiments were analyzed.

**Table 1 ijms-24-00502-t001:** Compounds selected for further experiments based on RGA results.

Compound	AhR Activation (24 h)		AR Inactivation (24 h)
	10 µM	100 µM	100 µM
3MI	2.6 fold	1.2 fold	decrease to 13.5%
4MI	4.5 fold	6.3 fold	decrease to 43.1%
1,3DMI	3.0 fold	2.7 fold	decrease to 63.5%
2,3DMI	3.9 fold	3.1 fold	decrease to 82.8%
2,3,7TMI	3.5 fold	3.5 fold	decrease to 63.5%
4,6DMI	2.6 fold	3.5 fold	decrease to 70.9%
5,6DMI	1.9 fold	2.6 fold	decrease to 55.5%
7MeO4MI	4.5 fold	4.7 fold	decrease to 70.8%

**Table 2 ijms-24-00502-t002:** Comparison of the effects of different compounds on AhR and AR receptors (↑ increase, ↓ decrease, - no effect; * only for certain compounds).

		AhR	AR
Tested Compound	Model		
**Icaritin**	**22Rv1**	↑ *CYP1A1* mRNA [24]; ↑ AhR protein level [24]	↓ protein levels [24]; ↓ *UBE2C, KLK3* mRNA [24]; ↓ recruitment to AR-target genes promoters [24]
	**LNCaP**	↑ *CYP1A1* mRNA [24]; ↑ AhR protein level [24]	↓ protein levels [24]; ↓ *UBE2C, KLK3* mRNA [24]; ↓ recruitment to AR-target genes promoters [24]
	**C4-2**		↓ protein levels [24]; ↓ *UBE2C, KLK3* mRNA [24]
**TCDD**	**22Rv1**	↑ *CYP1A1* mRNA	
	**LNCaP**	↑ *CYP1A1, CYP1B1* mRNA [20]; ↓ AhR protein level [20]	↓ AR protein level [20]; ↑ *KLK3* mRNA [20]; ↑ AR phosphorylation [20]
	**C4-2**	↑ *CYP1A1, CYP1B1* mRNA [20]; ↓ AhR protein level [20]	-AR protein level [20]
**Carbidopa**	**LNCaP**	↑ *CYP1A1* mRNA [25]; ↑ AhR protein level [25]	↓ AR protein level [25]; ↓ *PSA* mRNA [25]
**3 MI ***	**22Rv1** **(this study)**	↑ AhR activity (RGA); ↑ *CYP1A1* mRNA	↓ DHT-induced AR activity (RGA); ↓ AR-target genes mRNA; ↓ AR recruitment to *KLK3* promoter * ↓ AR protein level (dose-dependently)
**4 MI ***			
**1,3 DMI**			
**2,3 DMI**			
**2,3,7 TMI ***			
**4,6 DMI**			
**5,6 DMI**			
**7MeO4MI ***			

**Table 3 ijms-24-00502-t003:** List of used indolic compounds and abbreviations.

Compound	Abbreviation	Purity
1-methylindole	1MI	≥97%
2-methylindole	2MI	98%
3-methylindole	3MI	98%
4-methylindole	4MI	98%
5-methylindole	5MI	99%
6-methylindole	6MI	97%
7-methylindole	7MI	97%
1,2-methylindole	1,2DMI	99%
1,3-methylindole	1,3DMI	95%
2,3-methylindole	2,3DMI	≥97%
2,5-methylindole	2,5DMI	97%
2,3,3-trimethylindolenine	2,3,3TMI	98%
2,3,7-trimethylindole	2,3,7TMI	≥97%
4-methoxyindole	4MeO	99%
5-methoxyindole	5MeO	99%
6-methoxyindole	6MeO	98%
7-methoxyindole	7MeO	≥97%
4,6-dimethoxyindole	4,6DMI	≥98%
5,6-dimethoxyindole	5,6DMI	99%
4-methoxy-1-methylindole	4MeO1MI	98%
5-methoxy-2-methylindole	5MeO2MI	99%
7-methoxy-4-methylindole	7MeO4MI	95%

**Table 4 ijms-24-00502-t004:** Sequences of used primers and probes.

Target Gene	Sequence
*AhR exon 5*	f: 5′ TGAATTTCAGCGTCAGCTACA 3′
	r: 5′ AACAGACTACTGTCTGGGGGA 3′
*AhRR*	f: 5′GAGATGAAAATGAGGAGCGC 3′
	r: 5′TTTTACTTTTGCATCCGCGG 3′
	p: 5′[6FAM]AAACCCAGAGCAGACACCGCAGCCA[OQA] 3′
	f: 5′TGTGTCAAAAGCGAAATGGG 3′
*AR-fl*	r: 5′TTCATCTCCACAGATCAGGC 3′
	p: 5′[6FAM]TGCGTTTGGAGACTGCCAGGGACCA[OQA] 3′
*AR-v7*	f: 5′GAAATGTTAGAAGCAGGGATGACT 3′
	r: 5′GGTCATTTTGAGATGCTTGCAA 3′
	f: 5′GGAAGTGTATCGGTGAGACC 3′
*CYP1A1*	r: 5′CATAGATGGGGGTCATGTCC 3′
	p: 5′[6FAM]GCAACGGGTGGAATTCAGCGTGCCA[OQA] 3′
	f: 5′TCCAAGACTCAGATGATGCC 3′
*FKBP5*	r: 5′GGCACCCTGTAGTTATTTGC 3′
	p: 5′[6FAM]AAGTGTGTGTGGGGAGGGGAAGGGT[OQA] 3′
	f: 5′GAAGGAAATGAATGGGCAGC 3′
*GAPDH*	r: 5′TCTAGGAAAAGCATCACCCG 3′
	p: 5′[6FAM]ACTAACCCTGCGCTCCTGCCTCGAT[OQA] 3′
	f: 5′ACTGCATCAGGAACAAAAGC 3′
*KLK3*	r: 5′GGAGGCTCATATCGTAGAGC 3′
	p: 5′[6FAM]TGGGTCGGCACAGCCTGTTTCATCC[OQA] 3′
	f: 5′CCACAGTGAAGTTCCTCACG 3′
*UBE2C*	r: 5′GTTGGGTTCTCCTAGAAGGC 3′
	p: 5′[6FAM]ACCCCAACGTGGACACCCAGGGTAA[OQA] 3′

**Table 5 ijms-24-00502-t005:** RT-qPCR reaction setting for Taq-Man.

Detection Format	Mono Color Hydrolysis Probe/UPL Probe		
Reaction Volume	10 µL		
Program	Temperature	Time	Number of Cycles
Pre-incubation	95 °C	20 s	1
Amplification	95 °C	5 s	45
	58 °C	30 s	
Cooling	40 °C	30 s	1

**Table 6 ijms-24-00502-t006:** List of used antibodies.

Antibody	Type	Manufacturer	Dilution
**Primary antibody**			
β-actin	mouse monoclonal	Santa cruz Biotechnology	1: 2000
AR	mouse monoclonal	Santa cruz Biotechnology	1: 500
**Secondary antibody**			
Anti-mouse	IgG, HRP-linked	Santa cruz Biotechnology	1: 2000
Anti-rabbit	IgG, HRP-linked	Santa cruz Biotechnology	1: 2000

**Table 7 ijms-24-00502-t007:** ChIP RT-qPCR reaction setting.

Detection Format	SYBR Green I/HRM Dye		
Reaction Volume	10 µL		
Program	Temperature	Time	Number of Cycles
Pre-incubation	95 °C	10 min	1
Amplification	95 °C	15 s	40
	60 °C	1 min	
Melting curve	95 °C	5 s	1
	65 °C	1 min	
	97 °C	-	
Cooling	40 °C	10 s	1

## Data Availability

The data presented in this study are available online in this article.

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
