# Peer review of "The Impact of Indoles Activating the Aryl Hydrocarbon Receptor on Androgen Receptor Activity in the 22Rv1 Prostate Cancer Cell Line"

_ijms, 2022, doi:10.3390/ijms24010502_

Round 1
Reviewer 1 Report
In this paper, Zgarbová et al proposed that some indoles could activate AhR possess AR inhibiting activity, and are related to downregulation of AR expression rather than to AR degradation only. And it should not connect AhR activation with AR activity suppression in 22Rv1 cells directly during current studies. The data was sufficient in this paper, and the logic line was clear. But I suggest that authors should provide key structures of indoles discussed in this paper, which I also understood as mentioned in their previous studies. And they should shorten the abstract section to make key points clear, for example, how many indoles were studied? and which were active? Also, authors should isolate a separate paragraph for describing the conclusions in the end of the paper.
Author Response
Reviewer #1:
In this paper, Zgarbová et al proposed that some indoles could activate AhR possess AR inhibiting activity, and are related to downregulation of AR expression rather than to AR degradation only. And it should not connect AhR activation with AR activity suppression in 22Rv1 cells directly during current studies. The data was sufficient in this paper, and the logic line was clear. But I suggest that authors should provide key structures of indoles discussed in this paper, which I also understood as mentioned in their previous studies. And they should shorten the abstract section to make key points clear, for example, how many indoles were studied? And which were active? Also, authors should isolate a separate paragraph for describing the conclusions in the end of the paper.
Response: We are very grateful to the opponent for his/her comments and insights. An overview table with structural formulas of individual indolic compounds was added as a part of Supplement.
Moreover, the abstract section was edited and the concluding paragraph was added to the end of the manuscript.

Reviewer 2 Report
In this manuscript, the authors examined the effect of several indole compounds on AHR and AR in prostate cancer cells. Although this paper contains some interesting findings, the results are too preliminary to draw a conclusion.
1. Fig. 4. The quality of ChIP experiments is very poor. There are large variations among experiments. The authors should solve the technical problems.
2. Fig. 6. The effect of TCDD or 4MI on CYP1A1 expression was not abolished in AhR KO cells. Is it sure that AhR is deleted in these cells? AHR genotype and expression of AhR should be examined, and AHR deletion should be confirmed.
3. The effect of AHR deletion on CYP1A1 expression in cells treated with TCDD or 4MI without DHT combination should be examined.
4. The effect of AHR deletion on indole-induced suppression of AR mRNA and protein expression should be examined.
5. The effect of indoles on cell proliferation should be evaluated. AHR dependency should also be examined.
Author Response
Reviewer #2:
- 4. The quality of ChIP experiments is very poor. There are large variations among experiments. The authors should solve the technical problems.
Response: We thank the reviewer for this comment, but we do not strictly share his/her opinion. Despite the certain degree of variability among experiments, there is a trend in which the positive control always enriched AR on the KLK3 promoter and the presence of tested compounds decreased the enrichment. However, in order to strengthen the power of our data, we performed one more experiment (which confirmed the trend, we stand behind) and for higher clarity we expressed the data as % of positive control, DHT. This transformation highlighted the trend, which was less noticeable when each experiment was present separately.
- 6. The effect of TCDD or 4MI on CYP1A1 expression was not abolished in AhR KO cells. Is it sure that AhR is deleted in these cells? AHR genotype and expression of AhR should be examined, and AHR deletion should be confirmed.
Response: As mentioned in the manuscript, 22Rv1 were only transiently transfected with the AhR CRISPR/Cas9 KO (sc-400297). This plasmid is specific to the sequence in exon 5, which codes DNA-binding domain. We designed custom-made primers that are specific for this exact region and measured the expression of this AhR exon 5 by RT-qPCR. The results obtained were added to the Figure 6. These results show a decrease in the AhR mRNA level in each knockout experiment. However, since it was a transient transfection, the decrease varied among individual experiments and reached of 20 to 50 % compared to WT cells.
Due to the fact of transient transfection and the presence of mixed population (AhR-WT and AhR-KO), the effect of TCDD and 4MI on CYP1A1 expression was not fully abolished in AhR-KO cells. However, a certain level of decrease in CYP1A1 induction in AhR-KO cells, as a reflection of partial AhR knockout, is observable.
- The effect of AHR deletion on CYP1A1 expression in cells treated with TCDD or 4MI without DHT combination should be examined.
Response: Requested expression was measured and added to the manuscript. However, a decrease in CYP1A1 induction reflects the AhR knockout in mixed population, as explained in previous point.
- The effect of AHR deletion on indole-induced suppression of AR mRNA and protein expression should be examined.
Response: The result of AhR deletion on indole-induced suppression of AR mRNA was added to the Supplement.
We also repeatedly tried to determine the protein levels of AhR and AR in CRISPR/Cas9 AhR KO 22Rv1 cells. Since the only samples we had, were isolated for mRNA detection by TRI-Reagent method, we were unable to get sufficient protein amount for western blot detection. We are aware that protein analysis of these samples is the missing link but within the time provided for revision we were not able to perform additional experiments due to the CRISPR/Cas9 plasmid delivery problems.
- The effect of indoles on cell proliferation should be evaluated. AHR dependency should also be examined.
Response: As already mentioned in the manuscript (2.2 The effects of indoles on AhR and AR transcription activity), the Crystal violet proliferation assay was performed for all tested indoles on 22Rv1 cells (Supplementary figure 2).
Despite the fact, that the effects of indoles on cell proliferation via AhR dependency would bring a significant contribution to the topic, we were not able to perform such experiments due to the problems with CRISPR/Cas9 plasmid delivery within time provided for revision. Since these assays are less sensitive than PCR, it would require to have fully knockout AhR cell line. We worked with transiently transfected only.

Reviewer 3 Report
The manuscript (ijms-2049810), The Impact of Indoles Activating Aryl Hydrocarbon Receptor on Androgen Receptor Activity in 22Rv1 Prostate Cancer Cell Line, shows the interesting results in the manuscript. Authors have presented quite comprehensive analysis here. One minor comment would like to provide here is that -
1. Authors should provide a simple benchmark Table to summarize and compare the current research work and other works. What would be the key metrics in the current study to improve or further understand from previous research work. This would be quite helpful for this referee and potential readers to get further insights of the research impact and contribution in this work as compared to other works.
2. Did authors have the variability data analysis in Figure 4 and Figure 6? As compared to other plots, Figure 4 and Figure 6 did not have the variability data analysis, why?
Due to the above comments, this referee would like to put the manuscript status as "Minor Revision" in the current phase.
Author Response
Reviewer #3:
- Authors should provide a simple benchmark Table to summarize and compare the current research work and other works. What would be the key metrics in the current study to improve or further understand from previous research work. This would be quite helpful for this referee and potential readers to get further insights of the research impact and contribution in this work as compared to other works.
Response: We thank the reviews for his/her suggestion of manuscript improvement. The benchmark table was added to the manuscript.
- Did authors have the variability data analysis in Figure 4 and Figure 6? As compared to other plots, Figure 4 and Figure 6 did not have the variability data analysis, why?
Response: We do apologize, we filled the figures with variability as requested.

Round 2
Reviewer 2 Report
The authors added the required data. All comments have been addressed.